# Combinatorial Optimization for Panoptic Segmentation: A Fully Differentiable Approach

**Ahmed Abbas**  **Paul Swoboda**

MPI for Informatics
Saarland Informatics Campus

## Abstract

We propose a fully differentiable architecture for simultaneous semantic and instance segmentation (a.k.a. panoptic segmentation) consisting of a convolutional neural network and an asymmetric multiway cut problem solver. The latter solves a combinatorial optimization problem that elegantly incorporates semantic and boundary predictions to produce a panoptic labeling. Our formulation allows to directly maximize a smooth surrogate of the panoptic quality metric by backpropagating the gradient through the optimization problem. Experimental evaluation shows improvement by backpropagating through the optimization problem w.r.t. comparable approaches on Cityscapes and COCO datasets. Overall, our approach of combinatorial optimization for panoptic segmentation (COPS) shows the utility of using optimization in tandem with deep learning in a challenging large scale real-world problem and showcases benefits and insights into training such an architecture.

## 1  Introduction

Panoptic segmentation is the task of simultaneously segmenting different semantic classes and instances of the same class [37]. Panoptic segmentation is challenging since neural networks (NN) may produce conflicting predictions (i.e. boundaries separating instances that are not closed contours, instance voting schemes with multiple maxima per instance etc.). Therefore most approaches combine NNs with a post-processing step to compute a final panoptic segmentation that resolves the conflicting evidence produced by NNs. In general, joint training of NNs with post-processing algorithms is an active research area. In our work we propose a fully differentiable approach for panoptic segmentation, our post-processing being a combinatorial optimization problem.

In this work we pursue the bottom-up approach building segmentations directly from pixels and combine CNNs with the asymmetric multiway cut problem (AMWC) [42]. The latter is an elegant combinatorial optimization problem that combines semantic and affinity predictions and directly produces a panoptic labeling. We train CNN and AMWC jointly so that the supervisory signal for training the CNN is influenced by the computations of the combinatorial optimization stage. The loss we propose to use for this training differs from common lower-level CNN losses and is a smooth surrogate closely corresponding to the final panoptic quality metric [37]. We show in this work how our conceptual contributions, i.e. using AMWC as a differentiable module and training on surrogate panoptic quality loss can be made to work together and yield performance improvements.

The general idea of combining optimization and neural networks and train them jointly has recently enjoyed resurgent interest. The fundamental problem for the specific task of combinatorial optimization is that the output of combinatorial problem is 0–1 valued, hence the loss landscape becomes piecewise constant and simply differentiating through a solver is not possible anymore. Several methods have been proposed to address this problem [10, 24, 26, 31, 56, 64]. To our knowledge our work is the first to utilize the perturbation techniques [24, 64] on a large-scale setting with scalable

35th Conference on Neural Information Processing Systems (NeurIPS 2021).

but suboptimal heuristic solvers. We give evidence that training works in this setting and gives performance benefits. To this end we propose a robust extension of the backpropagation technique [64] that gives better empirical convergence.

Our architecture is inspired by [14, 16] and consists of a ResNet-50 backbone, a semantic segmentation branch for computing class costs and an affinity branch for boundary predictions. Semantic and affinity costs are taken as input by the AMWC solver that returns a panoptic labeling. We first pre-train semantic and affinity branches with simple cross-entropy losses obtaining a strong baseline that achieves a performance similar or better than other bottom-up approaches [16, 27, 69]. We finetune subsequently with the AMWC solver and the panoptic surrogate loss via our new robust backpropagation approach and show further performance improvements.

Current state-of-the-art approaches use very large networks (e.g. Max-DeepLab [65] uses transformers containing more parameters than a ResNet-101). This might lead to the impression that advances in panoptic segmentation require deeper and more sophisticated architecture. We show that our simpler model can be significantly improved by a fully differentiable approach and argue that simpler models have not yet reached their full potential. Also, our simpler architecture allows for a more controlled setting and makes it easier to identify crucial components and measure to which extent performance improvements can be achieved.

### Contributions

**Optimization for segmentation:** We propose AMWC [42] as an expressive and tractable combinatorial optimization formulation to be used in an fully differentiable architecture for panoptic segmentation. We also propose a scalable heuristic for its solution.

**Panoptic loss surrogate:** We propose a surrogate loss function that approximates the panoptic loss metric and can be used in our training setup.

**Backpropagation:** We give an extension of the perturbation technique [64] for backpropagating gradients through combinatorial solvers, improving training with suboptimal heuristic solvers.

**Experimental validation:** We conduct experiments on Cityscapes [19] and COCO [47] and show the benefits of fully differentiable training against comparable approaches.

Our code is available at `https://github.com/aabbas90/COPS`.

## 2 Related Work

### 2.1 Panoptic segmentation

We categorize panoptic segmentation approaches into three categories: (i) bottom-up methods predict information on the pixel-level and then use post-processing to produce a segmentation, (ii) top-down methods proceed by first identifying regions of interest (ROI) and subsequently basing segmentation on them and (iii) hybrid methods combine bottom-up and top-down ideas. For a general overview of recent segmentation methods we refer to [51]. Here we will restrict to panoptic segmentation tasks.

**Top-down:** Recent works include [12, 36, 37, 43, 52, 58, 59, 73, 75]. This principle has also been used with weak supervision [45]. As a drawback, top-down approaches use ROIs which are mostly axis-aligned and so they can be in-efficient for scenarios containing deformable objects [63].

**Bottom-up:** Panoptic-DeepLab [16] based on [74] proposes a single-stage neural network architecture which combines instance center of mass scores with semantic segmentation to compute panoptic segmentation. They use post-processing similar to Hough-voting [6], obtaining great results and reducing the gap to top-down approaches. Subsequently, Axial-DeepLab [66] made improvements using an attention mechanism to enlarge the receptive field using the post-processing scheme of [74].

The methods SSAP [27] and SMW [69] are most similar to our as they also use semantic and affinity scores with a graph partitioning algorithm. SMW [69] additionally uses Mask-RCNN [29] and SSAP solves multiple graph partitioning problems in coarse-to-fine manner. Older works such as [38, 48] use graph partitioning schemes but only for the instance segmentation task.

**Hybrid:** The approaches [46, 69] use both bottom-up (affinity scores) and top-down (bounding boxes) sources of information. Conditional convolution [63] was used in [65]. Transformers are used in [12] and combined with Max-DeepLab in a sophisticated architecture, achieving remarkable results. They used a surrogate for the panoptic quality metric along with an instance discrimination

loss similar to [71]. However, Max-DeepLab imposes an upper bound on the maximum number of instances in an image and requires thresholding low confidence predictions.

In summary, bottom-up methods are generally simpler than top-down ones and require fewer hyper-parameters. However, they lack global context and are generally outperformed by top-down approaches. As a solution Axial-DeepLab [66] reduce this gap by incorporating long range context.

Almost all of the above-mentioned approaches use multiple loss functions (see [33] for a possible solution), need thresholds for getting rid of low confidence predictions or assume an upper bound on the number of instances and therefore require hyperparameter tuning. To achieve end-to-end training, approaches of [12, 46, 65] design mechanisms embedded in the NNs which can compute panoptic segmentations directly but still have test-time hyperparameters (such as maximum number of instances, probability thresholding) and need more complicated architectures. Except for the above works, other approaches delegate this task to a post-processing module which does not participate in training. The motivation of our work is based on prioritizing ease-of-use and simplicity. Therefore we have chosen a bottom-up approach and propose a fully differentiable method for training with only one loss and no ad-hoc downstream refinements of the segmentation.

## 2.2 Algorithms as a layer in neural networks

Recently there has been great interest in training neural networks with additional layers for problem-specific constraints and prior knowledge. The works [28, 41] provide an extensive survey and insights. An excellent overview of multiple approaches for learning graphical model parameters is given in [25]. The focus of our work is on using an optimization problem as a layer in neural networks. Hence, we will mainly cover approaches for this scenario. They can be categorized as follows:

**Unrolling:** For training NNs together with cheap and differentiable iterative algorithms (or for algorithms that can be made differentiable e.g. by smoothing), straightforwardly computing gradients is the most simple approach. This has been done for K-means [68] bipartite matching [76], conditional random fields [4, 24, 62, 77], non-linear diffusion for image restoration [15] and ranking and sorting [21]. The interesting study [18] shows that under some stability conditions backpropagation through the last few steps of iterative procedures is enough to get good estimates of gradients.

**Implicit Function Theorem:** In case solutions satisfy fixed point conditions (e.g. KKT conditions) the implicit function theorem can be used to compute gradients. This was done for quadratic programs in [2], embedding MaxSAT in neural networks [67], a large class of convex optimization problems [1], smoothed top-k selection via optimal transport [72] and deep equilibrium models [5].

**Problem-specific methods:** Specialized approaches for backpropagating for specific problems were investigated for submodularity [23] (e.g. using a graph-cut layer), belief propagation [40], dynamic programming [50], markov random fields [13, 39] and nearest neighbor selection [57].

**Gradient Estimation by Perturbation:** Perturbing the objective of an optimization problem for learning has been proposed in [9, 44, 53] for graphical model parameters. In [10, 20, 55] perturbation is used in the forward pass to get a differentiable estimate of the solution. Perturbing the objective in the direction of loss decrease has been proposed in [24] for backpropagating through graphical model inference, in [49] to estimate gradients through a structured loss and in [64] to backpropagate through combinatorial optimization problems. The latter was used for ranking [60] and graph matching [61].

## 3 Method

Our architecture shown in Figure 2 is comprised of two stages: (i) a CNN to compute semantic class and affinities for boundary predictions followed by (ii) an AMWC optimization layer producing the final panoptic labeling. We describe below our CNN architecture, the AMWC problem and finally the approach for backpropagting through the AMWC solver to optimize panoptic surrogate loss.

### 3.1 CNN Architecture

Our CNN architecture (see Figure 2) is comprised of the following parts: a shared ResNet-50 backbone pre-trained on ImageNet [22] producing feature maps for the subsequent semantic and affinity branch. Our CNN architecture corresponds to Panoptic-Deeplab [16] with the exception of a modified instance segmentation branch due to different post-processing (Hough voting for vs.

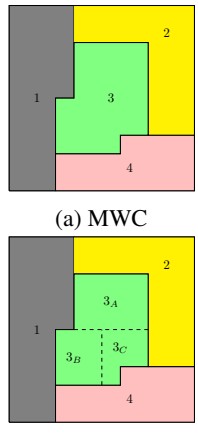

(a) MWC

(b) AMWC

Figure 1: Exemplary MWC and AMWC problems with 4 classes ($K = 4$). MWC is a special case of AMWC when $P = \varnothing$. For $P = \{3\}$ we get an AMWC problem where class 3 is partitioned into subclusters (instances) $3_A$, $3_B$ and $3_C$.

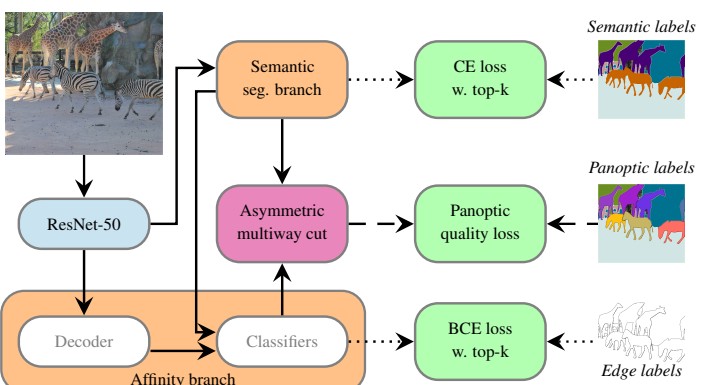

Figure 2: Overview of our architecture: Image features computed through a ResNet-50 backbone [30] are fed into a semantic segmentation branch to predict class scores and to an affinity branch to predict object boundaries. Costs from both branches are used in the AMWC solver for computing a panoptic labeling. Pre-training of the semantic and affinity branch is done with top-k cross-entropy losses [74] (dotted arrows). For backpropagation through AMWC solver we use the panoptic quality loss (dashed arrows). The computation flow marked by solid lines is for panoptic segmentation, dotted arrow for pre-training and dashed arrows for fully differentiable training.

AMWC in our work). We also use DeepLabv3+ [14] decoders for both semantic and affinity branch similar to [16]. This allows for a fair comparison with [16].

**Affinity predictor:** The affinity branch predicts for given pairs of pixels whether they belong to the same instance. It takes two sources of inputs: (i) features from the affinity decoder and (ii) semantic segmentation costs which makes finding boundaries between different classes easier. Gradients of segmentation costs computed from affinity predictors are not backpropagated during training to preclude the affinity branch from influencing the semantic branch.

We take horizontal and vertical edges at varying distances $d$. For COCO we use $d \in \{1, 4, 16, 32, 64\}$ and for Cityscapes $d \in \{1, 4, 16, 32, 64, 128\}$. For each $d$ all corresponding edges are sampled and affinity scores are computed by a dedicated predictor for each distance. For long range edges with $d > 1$ we compute edge features by taking the difference of affinity features of the edge endpoints before sending them to the predictor. This helps in capturing long-range context. Additional architectural details can be found in the appendix.

## 3.2 (Asymmetric) Multiway Cut

Multiway cut (MWC) [11] is a combinatorial optimization problem for graph partitioning defined on a graph. In MWC a pre-defined number of classes is given and each node is assigned to one. The cost of a class assignment is given by node and edge affinity costs that give the preference of a node belonging to a certain class and endpoints of the edge to belong to the same class respectively. Hence, the multiway cut can be straightforwardly used to formulate semantic segmentation, each MWC class corresponding to a semantic class.

The AMWC problem was introduced in [42] as an extension of MWC. AMWC additionally allows to subdivide some classes into an arbitrary number of sub-clusters. This allows to model segmenting a given semantic class into multiple instances for panoptic segmentation.

Mathematically, MWC and AMWC are defined on a graph $G = (V, E)$ together with edge weights $c_E : E \rightarrow \mathbb{R}$ and node costs $c_V : V \times \{1, \ldots, K\} \rightarrow \mathbb{R}$, where $K$ is the number of classes. The edge affinities $c_E$ indicate preference of edge endpoints to belong to the same cluster, while the node costs $c_V$ indicate preference of assigning nodes to classes. A set $P \subseteq [K]$ contains classes that can be partitioned. For MWC we have $P = \varnothing$ while for AMWC $P \subseteq [K]$. Let B be the set of valid

boundaries, i.e. edge indicator vectors of partitions of $V$

$$\mathrm{B} = \{y : E \to \{0,1\} : \exists\, C_1 \dot{\cup} \ldots \dot{\cup}\, C_M \text{ s.t. } y(ij) = 1 \Leftrightarrow \exists\, l \neq l' \text{ and } i \in C_l, j \in C_{l'}\} \quad (1)$$

where the number of clusters $M$ is arbitrary and $\dot{\cup}$ is the disjoint union. The MWC and AMWC optimization problems can be written as

$$\begin{array}{cl} \min\limits_{x:V\to\{1,\ldots,K\},y\in\mathrm{B}} & \sum_{i\in V} c_V(i,x(i)) + \sum_{ij\in E} c_E(ij)\cdot y(ij) \\ \text{s.t.} & y(ij) = 0, \text{ if } x(i) = x(j) \notin P \\ & y(ij) = 1, \text{ if } x(i) \neq x(j) \end{array} \quad (2)$$

The above constraints stipulate that $y$ produces a valid clustering of the graph compatible with the node labeling $x$, i.e. boundaries implied by $y$ align with class boundaries defined by $x$ and non-partitionable classes not in $P$ do not possess internal boundaries. The AMWC can be thought of as a special case of InstanceCut [38] that has class-dependent edge affinities, which, however, makes it less scalable. Illustrations of MWC and AMWC are given in Figure 1.

Given a feasible solution $(x,y)$ satisfying the constraints in (2), the panoptic labeling $z : V \to \{1,\ldots,J\}$ is computed by connected components w.r.t. $y$, i.e. $z(i) = z(j) \Leftrightarrow y(ij) = 0,\ \forall ij \in E$.

Optimization algorithms for efficiently computing possibly suboptimal solutions for (2) are given in the appendix. Note that, contrary to other approaches for panoptic segmentation such as [63, 65, 73] AMWC neither has an upper bound on the number of instances $M$ (which is automatically decided by the optimization problem) nor suffers from computational bottlenecks in this regard. It also does not require thresholding to get rid of low confidence predictions.

### 3.3 Fully differentiable training

To train our architecture along with the AMWC solver we first introduce a new robust variant of the perturbation technique for backpropagation [64] which works well for our setting of a large-scale problem and suboptimal solver. Second, we introduce a smooth panoptic loss surrogate. Last, we show how to backpropagate gradients for the panoptic loss surrogate through a MWC layer.

#### 3.3.1 Robust Perturbation for Backpropagation:

The fundamental difficulty of backpropagating through a combinatorial optimization problem is that the loss landscape is piecewise constant, since the output of the combinatorial problem is integer valued. To handle this difficulty, generally applicable perturbation techniques [7, 24, 31, 49, 64] have been proposed. They work by taking finite differences of solutions with perturbations of the original problem. The work [64] interprets this as creating a continuous interpolation of the non-continuous original loss landscape.

The second difficulty is that, due to large size and NP-hardness of AMWC, we use a heuristic suboptimal solver that does not in general deliver optimal solutions. Therefore, we propose a multi-scale extension of [64] for increased robustness that works well in our setting.

Assume a binary integer linear optimization layer $\mathcal{W}$ takes a cost vector $c$ as input from a neural network i.e. $\mathcal{W} : \mathbb{R}^n \to \{0,1\}^n, c \mapsto \arg\min_{x\in\mathcal{S}}\langle c,x\rangle$ where $\mathcal{S} \subset \{0,1\}^n$ is the set of constraints. Afterwards the minimizer of $\mathcal{W}$ is fed into a loss function $L : \{0,1\}^n \to \mathbb{R}$. For backpropagation we need to compute the gradient $\frac{\partial(L\circ\mathcal{W})}{\partial c}$, where $L \circ \mathcal{W}$ is the composition of $L$ and $\mathcal{W}$. Since, this gradient is zero almost everywhere a continuous interpolation $(L \circ \mathcal{W})_\lambda$ is proposed in [64] where $\lambda > 0$ is an interpolation range. The gradient w.r.t. the interpolation is computed by perturbation of the cost vector $c$ by incoming gradient as follows

$$\frac{\partial(L\circ\mathcal{W})_\lambda}{\partial c} = \frac{1}{\lambda}\Big[\mathcal{W}(c + \lambda\nabla L(\mathcal{W}(c))) - \mathcal{W}(c)\Big] \quad (3)$$

while [64] report that a large interval of interpolation ranges $\lambda$ work well on their test problems with optimal solvers, we have not been able to confirm this for our suboptimal heuristic that only gives approximately good solutions to $\mathcal{W}$. Therefore, we propose to use a multi-scale loss and its gradient

$$(L\circ\mathcal{W})_{avg} := \frac{1}{N}\sum_{i=1}^{N}(L\circ\mathcal{W})_{\lambda_i}, \qquad \frac{\partial(L\circ\mathcal{W})_{avg}}{\partial c} = \frac{1}{N}\sum_{i=1}^{N}\frac{\partial(L\circ\mathcal{W})_{\lambda_i}}{\partial c} \quad (4)$$

where $\lambda_i$ are sampled uniformly in an interval. While the robust backpropagation formula (4) needs multiple calls to the optimization oracle $\mathcal{W}$, they can be computed in parallel. In practice the computation time for a backward pass will hence not increase.

### 3.3.2 Panoptic Quality Surrogate Loss:

Panoptic quality (PQ) [37] is a size-invariant evaluation metric defined between a set of predicted masks and ground-truth masks for each semantic class $l \in [K]$. For each class, it requires to match predicted and object masks to each other w.r.t intersection-over-union (IoU) since instance labels are permutation invariant. A pair of predicted and ground truth binary masks $p$ and $g$ of the same class $l$ is matched (i.e. true-positive) if $IoU(p, g) \geq 0.5$. We write $(p, g) \in TP_l$. For the unmatched masks, each prediction (ground-truth) is marked as false positive $FP_l$ (false negative $FN_l$). Since at most one match exists per ground truth mask, this matching process is well-defined [37]. The PQ metric is defined as the mean of class specific PQ scores

$$PQ_l = \frac{\sum_{(p,g)\in TP_l} IoU(p, g)}{|TP_l| + 0.5(|FP_l| + |FN_l|)} \tag{5}$$

Note that the PQ score (5) can be arbitrarily low just by the presence of small sized false predictions [16, 58, 73]. A common practice to avoid such issue is to reject small predictions before computing the PQ score with some dataset specific size thresholds, before evaluation. However, this rejection mechanism is not incorporated during training.

The PQ metric (5) cannot be straightforwardly used for training due to the discontinuity of the hard threshold based matching and the rejection mechanism. Therefore we replace the hard threshold matching process for each class $l$ by computing correspondences via a maximum weighted bipartite matching with $IoU$ as weights. The corresponding matches are $\overline{TP}_l$, the unmatched prediction masks $\overline{FP}_l$ and the unmatched ground truth masks $\overline{FN}_l$. The hard thresholding is smoothed via soft thresholding function $h(u) = \frac{u^4}{u^4+(1-u)^4}$ centered around $0.5$. The small prediction rejection mechanism for mask $p$ is smoothed via $\sigma_l(p) = [1 + \exp(-0.1(1^T p - t_l))]^{-1}$ centered at area threshold $t_l$ for class $l$. The overall surrogate PQ for class $l$ is

$$\overline{PQ}_l = \frac{\sum_{(p,g)\in \overline{TP}_l} h(IoU(p, g))\, \sigma_l(p)\, IoU(p, g)}{\sum_{(p,g)\in \overline{TP}_l} h(IoU(p, g))\, \sigma_l(p) + 0.5\{\sum_{p\in \overline{FP}_l} \sigma_l(p) + |\overline{FN}_l|\}} \tag{6}$$

where the term $h(IoU(p, g))$ models the probability of a predicted mask $p$ being true positive.

### 3.3.3 Transformation to Multiway Cut:

In order to directly train with the panoptic loss surrogate (6) via the backpropagation formula (4) we propose a transformation of the AMWC problem to a lifted MWC problem in the backward pass for computing gradients. The AMWC optimization oracle $\mathcal{W}$ can be written as

$$(x^*, y^*, z^*) = \arg\min_{x,y,z} \langle c_V, x \rangle + \langle c_E, y \rangle \tag{7}$$
$$\text{s.t.} \quad z(i) = z(j), \text{ if } y(ij) = 0$$
$$z(i) \neq z(j), \text{ if } y(ij) = 1$$
$$(x, y) \in \mathcal{S}, z \in \mathbb{Z}_+$$

where $\mathcal{S}$ describes the constraint listed in (2) and the loss is calculated w.r.t panoptic labels $z^*$ i.e. $\mathcal{W}(c_V, c_E) = z^*$. To compute the gradients as per (3) we need to perturb the cost vector associated with $z$ in (7). However, AMWC only takes semantic costs and affinity costs as input not the panoptic costs. In other words, the gradient of (6) affects node costs of individual instances separately (i.e. they work on panoptic labels), but AMWC assumes node costs are equal for all instances of one semantic class (i.e. it works on class labels). Therefore we transform the AMWC problem into a lifted MWC problem that has a class for each panoptic label in the ground truth. This allows to optimize directly in panoptic label space and compute a gradient w.r.t semantic and affinity costs which can then be backpropagated to corresponding branches.

**Algorithm 1:** BACKWARD PASS

**Input** : $\frac{\partial L}{\partial z}, c_V, c_E$, solution $x, y, m, \lambda$
**Output** : $\frac{\partial L}{\partial c_V}, \frac{\partial L}{\partial c_E}$

**1** Transform node costs to panoptic and perturb:
$$c'_V(l) = c_V(m(l)) + \lambda \frac{\partial L}{\partial z}(l), \ \forall l \in [J]$$
**2** Multiway cut on panoptic label space:
$$(z_p, y_p) = \text{MWC}(c'_V, c_E)$$
**3** Perturbed class labels:
$$x_p(i) = m(z_p(i)), \forall i \in V$$
**4** Compute node cost gradients: $\frac{\partial L}{\partial c_V} = \frac{1}{\lambda}(x_p - x)$

**5** Compute edge cost gradients: $\frac{\partial L}{\partial c_E} = \frac{1}{\lambda}(y_p - y)$
**6** **return** $\frac{\partial L}{\partial c_V}, \frac{\partial L}{\partial c_E}$

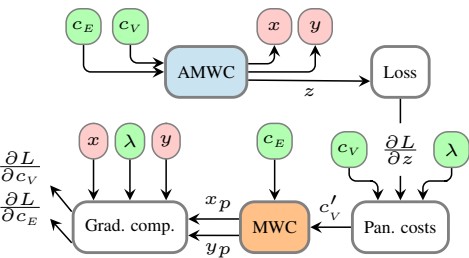

Figure 3: Gradient computation for $c_V, c_E$ for fully differentiable learning: AMWC produces semantic, edge, panoptic labels $x, y, z$ resp. Perturbations of panoptic label costs $c'_V$ are computed and sent to the MWC solver together with the original edge costs $c_E$. Results are used to compute and return the gradients.

For the backward pass described in Algorithm 1 we define the following notation: Let $J$ be the number of classes for the lifted MWC problem and $m : [J] \rightarrow [K]$ the mapping from panoptic labels onto the corresponding semantic class. Algorithm 1 computes the gradient w.r.t. the simple backpropagation formula (3). For the robust backprop (4) the algorithm has to be called multiple times with the corresponding interpolation ranges $\lambda$. An illustration of the gradient computation is given in Figure 3.

Line 1 in Alg. 1 merges two sources of information i.e. preference of the loss $L$ on panoptic labels $z$ and current class costs $c_V$. Note that the edge costs $c_E$ are not perturbed. Afterwards, the perturbed panoptic labels $z_p$ are converted back to class labels $x_p$ on line 3 to compute the gradients. Ablation study w.r.t using simpler losses on the output of AMWC i.e. class labels $x$ and edge labels $y$ solver is shown in the appendix. Intuitively, applying a loss directly on edge labels does not work because small and large localization errors in edge labels are treated equally. This issue was also observed in [3] for 3D instance segmentation.

## 4 Experiments

All baselines are trained on NVIDIA Quadro RTX 8000 GPUs with $48GB$ memory each. For fully differentiable training we use one Tesla P40 with $24GB$ memory and a 32 core CPU to solve all AMWC problems in the batch in parallel.

### 4.1 Datasets

We train and evaluate COPS on the Cityscapes [19] and COCO [47] panoptic segmentation datasets. We test on the validation and test sets provided by the two datasets. For evaluation on the test set we do not use validation set for training.

**Cityscapes:** Contains traffic related images of resolution $1024 \times 2048$ where training, validation and testing splits have $2975, 500$, and $1525$ images for training, validation, and testing, respectively. It contains 8 'thing' and 11 'stuff' classes. During training we use random scale augmentation and crop to $512 \times 1024$ resolution as done in Panoptic-DeepLab. During evaluation the input images are sent at original resolution. The values of small segment rejection thresholds (used during both training and inference) are $200, 2048$ for 'thing'and 'stuff'class resp. Lastly, to handle larger occlusions we additionally use affinities at a distance of $128$.

**COCO:** Is more diverse and contains $118k, 5k$, and $20k$ images for training, validation, and testing, resp. The dataset has 80 'thing' and 53 'stuff' classes. During training random scale augmentation is also used with a crop size of $640 \times 640$ resolution as in [16]. The values of small segment rejection thresholds (used during both training and inference) are $200, 4096$ for 'thing'and 'stuff'class resp. During evaluation the input images are resized to $640 \times 640$ resolution.

## 4.2 Training

We closely follow the implementation of Panoptic-DeepLab in [70] (based on Pytorch [54]), use the provided ImageNet pre-trained ResNet-50 backbone and the same learning rate parameters for training our baseline model. The Adam optimizer [35] is used for all our experiments.

**Resolution:** The CNNs produce an output with $1/4$-th the resolution in every dimension w.r.t input images, similar to Panoptic-DeepLab. This reduced input size is maintained for AMWC (instead of upsampled) to reduce computation time during full training and evaluation. The panoptic labels computed by the AMWC solver are upsampled during evaluation. Since these labels are discrete, upsampling may misalign object boundaries and small ground-truth objects can potentially be missed as well. While this can put our method at a disadvantage, our full training scheme offsets this by achieving panoptic quality even better than the performance at finest resolution of comparable methods.

**Baseline pre-training:** We pre-train the CNN architecture as a baseline model and for achieving a good initialization for the subsequent fully differentiable training. This also allows us to measure the additional gain by full training. In pre-training we apply the weighted top-k cross-entropy loss [74] to each affinity predictor separately and also to the semantic segmentation branch. Since the main objective of the affinity classifier should be to predict instance boundaries we increase the loss by a factor of $4$ for edges where at least one endpoint belongs to a 'thing' class. Additionally, we also increase the semantic and affinity loss weights of small objects by a factor of $3$ following [16].

We train Cityscapes on one GPU with batch-size $12$ for $250k$ iterations, with initial learning rate $0.001$ and the decay strategies of Panoptic-DeepLab. Training takes around $8$ days. COCO is trained on four GPUs with a total batch-size of $48$ for $240k$ iterations using the same learning rate parameters as above. Training takes around $11$ days.

**Full training:** For training COPS through AMWC solver we use only the panoptic quality surrogate loss (6) and fine-tune the semantic and affinity classifiers along with the last layer of each semantic and affinity decoder. The ResNet50 backbone and all batch normalization parameters [32] are frozen. We train with batch size of $24$ until training loss convergences which amounts to $3000$ iterations for Cityscapes and $10000$ iterations for COCO. To approximate the gradient (4) we use relatively large values of $\lambda$ compared to [64] since in-exact optimization might not react to small perturbations correctly (for example the backward pass solution might not even be equal to the one from the forward pass for $\lambda \to 0$). We also observed more stable training curves for larger values of $N$ and use $N = 5$ in our experiments.

## 4.3 Results

We compare panoptic quality (in terms of percentage) on both testing $\text{PQ}^{\text{test}}$ and validation $\text{PQ}^{\text{val}}$ splits of Cityscapes and COCO datasets, see Table 1. For the testing splits evaluation requires submission to an online server. We also show performance on 'thing' classes $\text{PQ}_{\text{th}}$, and stuff classes $\text{PQ}_{\text{st}}$ separately. To allow a fair comparison, we restrict ourselves to results of competing approaches which are closest to our setting i.e., without test-time augmentation, similar number of parameters in the network, not utilizing other sources of training data etc. For an overall comparison, we also consider at least one state-of-the-art work from each other type of method (top-down, hybrid etc.).

First, our fully trained model improves by more than $3$ and $4$ points in panoptic quality for Cityscapes and COCO resp. in comparison to our baseline model. This is evidence our panoptic loss surrogate and training in conjunction with the combinatorial solver works. Especially, performance on the 'thing' classes improves which have internal boundaries. We argue this is mainly due to better training of the affinity branch, which benefits more from the AMWC supervisory signal. A sample qualitative comparison between baseline and fully trained model can be seen in Figure 9, where full training shows clear visible improvements. The methods SSAP [27], SMW [69] are closest to ours in-terms of the post-processing, and Panoptic-DeepLab in-terms of architecture resp. Our fully trained model outperforms SSAP even in a setting where SSAP uses test-time augmentation and a larger backbone. SMW reports results only on Cityscapes using two independent DeepLabV3+ models and a Mask-RCNN. We outperform it with our approach while still using a simpler model. While Panoptic-Deeplab outperforms our baseline model, our full training scheme outperforms it on both datasets.

Table 1: Results on Cityscapes (above) and COCO (below) on validation and testing splits. We divide the methods into two groups where lower half for each dataset contains the approaches which are comparable to COPS with bold numbers representing the best performance in this category. R-X: ResNet-X, X-71: Xception-71, †: Mask selection (e.g. by Mask-RCNN), *: Uses test-time augmentation. (-) Marks the results which are not reported for that setting.

| Method | Backbone | $PQ^{test}$ | $PQ_{th}^{test}$ | $PQ_{st}^{test}$ | $PQ^{val}$ | $PQ_{th}^{val}$ | $PQ_{st}^{val}$ |
|---|---|---|---|---|---|---|---|
| Cityscapes | | | | | | | |
| Axial-DL [66] | Axial-L | 62.7 | 53.4 | 69.5 | 63.9 | - | - |
| EfficientPS [52]† | Custom | - | - | - | 63.9 | 66.2 | 60.7 |
| Panoptic-DL [16] | X-71 | 60.7 | - | - | 63.0 | - | - |
| Unify. PS [46]† | R-50 | 61.0 | 52.7 | 67.1 | 61.4 | 54.7 | 66.3 |
| UPSNet [73]† | R-50 | - | - | - | 59.3 | 54.6 | 62.7 |
| Panoptic-FPN [36]† | R-101 | - | - | - | 58.1 | 52.0 | 62.5 |
| SSAP [27]* | R-101 | 58.9 | 48.4 | **66.5** | 61.1 | 55.0 | - |
| Panoptic-DL [16] | R-50 | 58.0 | - | - | 60.3 | 51.1 | 66.9 |
| SMW [69]† | Multiple | - | - | - | 59.3 | 50.6 | 65.7 |
| Unify. PS [46] | R-50 | - | - | - | 59.0 | 50.2 | 65.3 |
| SSAP [27] | R-50 | - | - | - | 56.6 | 49.2 | - |
| COPS baseline | R-50 | 56.7 | 46.0 | 64.5 | 58.5 | 48.3 | 66.0 |
| COPS full | R-50 | **60.0** | **51.8** | 65.9 | **62.1** | **55.1** | **67.2** |
| COCO | | | | | | | |
| Max-DeepLab [65] | MaX-S | 49 | 54 | 41.6 | - | - | - |
| Unify. PS [46]† | R-50 | 43.6 | 48.9 | 35.6 | 43.4 | 48.6 | 35.5 |
| UPSNet [73]† | R-50 | - | - | - | 42.5 | 48.5 | 33.4 |
| Axial-DL [66] | Axial-S | 42.2 | 46.5 | 35.7 | 41.8 | 46.1 | 35.2 |
| Panoptic-FPN [36]† | R-101 | 40.9 | 48.3 | 29.7 | 40.3 | 47.5 | 29.5 |
| Panoptic-DL [16] | X-71 | 38.8 | - | - | 39.7 | 43.9 | 33.2 |
| SSAP [27]* | R-101 | 36.9 | 40.1 | 32 | 36.5 | - | - |
| Panoptic-DL [16] | R-50 | 35.2 | - | - | 35.5 | 37.8 | 32.0 |
| COPS baseline | R-50 | 34.2 | 35.2 | 32.8 | 34.3 | 34.9 | 33.4 |
| COPS full | R-50 | **38.5** | **41.0** | **34.8** | **38.4** | **40.5** | **35.2** |

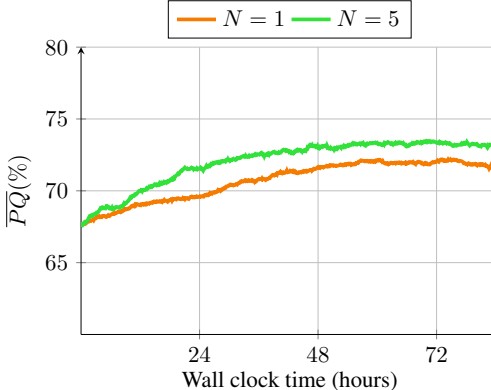
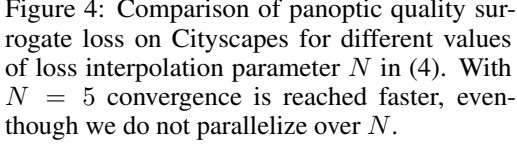

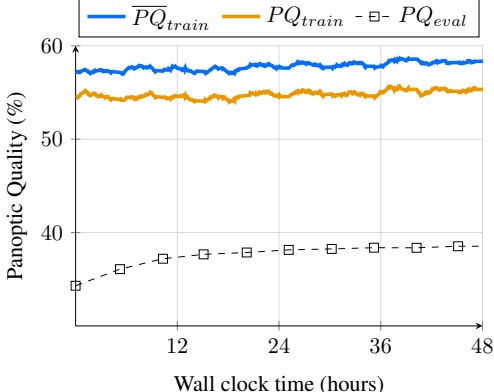

Figure 4: Comparison of panoptic quality surrogate loss on Cityscapes for different values of loss interpolation parameter $N$ in (4). With $N = 5$ convergence is reached faster, eventhough we do not parallelize over $N$.

Figure 5: Train, eval. logs on COCO dataset during fully differentiable training. $\overline{PQ}_{train}$ (6) and $PQ_{train}$ (5) are computed during training. $PQ_{eval}$ (5) is reported on the whole COCO validation set after every 1000 iterations.

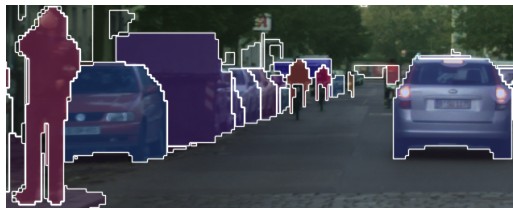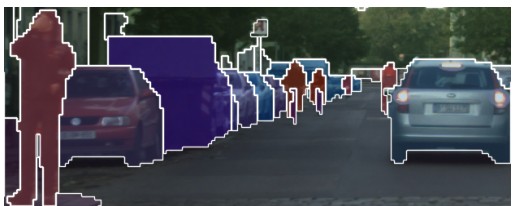

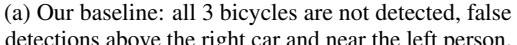

(a) Our baseline: all 3 bicycles are not detected, false detections above the right car and near the left person.

(b) After full training: better localization and bicycles are correctly detected

Figure 6: Comparison of panoptic labels on Cityscapes test set. (Best viewed digitally).

In Figure 4 we plot the PQ surrogate (6) during fully differentiable training using different numbers of interpolation parameter $N$ in (4). Our proposed improvement in the backpropagation scheme of [64] trains faster and achieves better panoptic quality. In Figure 5 we compare our differentiable PQ surrogate (6) with the exact PQ metric (5) during training. Note that PQ surrogate overestimates exact PQ because we smooth hard thresholding operators. Lastly, we see significant improvement in PQ on evaluation set already after only 24 hours of training with a batch-size of 24 (baseline training took 11 days with 48 batch-size).

## 4.4  Limitations

**Inference times:**  Although parallelization can be simply done during training, our approach lacks real-time performance during inference requiring around 2 seconds per image from Cityscapes and 0.3 seconds for COCO.

**Two stage training:**  Our training procedure two steps. First we pre-train the network using simpler losses and then finetune with panoptic quality surrogate loss by backpropagating through AMWC. We follow this approach due to computational efficiency, since the combinatorial part takes a significant amount of time. We hope that with better and faster AMWC solvers training can be converted to a single stage in the future. Moreover, we avoid finetuning the whole model with panoptic quality surrogate because IoU based metrics are not separable under expectations w.r.t. different images [8]. To get good estimates of the loss we therefore require larger batch sizes than for simpler losses used in pre-training. This restriction makes it difficult to train all layers due to GPU memory limitations. It would be interesting to train all parameters by backpropagation through the combinatorial solver and forego the need for pre-training possibly on applications with simpler losses and fast combinatorial solvers.

## 5  Conclusion

We have proposed a fully differentiable approach for panoptic segmentation incorporating a combinatorial optimization layer for post-processing and directly minimizing panoptic quality surrogate loss. Our choice has lead to a simple and elegant formulation with a minimal number of hyperparameters. We argue that learning through combinatorial optimization layers is possible and leads to improved performance even with simple and suboptimal solvers. However, backpropagation schemes should be suitably augmented for robustness in this case.

While our work suggests that combinatorial optimization is helpful in neural networks, most solvers (including the ones we used) are sequential and executed on CPU, which limits their applicability. For combinatorial optimization to become a more commonly used layer in neural networks, solvers must be designed that are inherently parallel and executable on GPUs.

## Broader Impact

This work introduces a new fully differentiable architecture for panoptic segmentation, a fundamental task in computer vision used in down-stream tasks. The broader impact of our work depends on the concrete downstream task.

## Acknowledgements

We would like to thank Michal Rolínek for his valuable suggestions regarding backpropagation through optimization problems.

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
