# A  Supplementary Material

## A.1  AMWC Heuristic

Algorithm 2 describes the heuristic we use for asymmetric multiway cut inspired by greedy additive edge contraction heuristic for multicut in [34]. The algorithm proceeds by initializing each vertex belonging to a separate cluster (panoptic label). Afterwards, most similar edges are merged in a greedy fashion until similarity becomes negative (Lines 4-6), where similarity for an edge is computed in accordance with its affinity cost as well as node costs (Lines 25-29). Whenever a merge operation is performed the corresponding edge is contracted and new edges can potentially be created (Lines 7-17). Afterwards, the clusters belonging to non-partitionable class (i.e stuff) are merged. For finding the $\arg\max$ efficiently in Line 4 we use priority-queue.

---

**Algorithm 2:** AMWC GREEDY EDGE CONTRACTION

**Input** : Graph $G = (V, E)$, node costs $c_V : V \times [K] \to \mathbb{R}$, edge costs $c_E : E \to \mathbb{R}$, partitionable classes $P \subseteq [K]$

**Output :** Clustering $C_1 \dot{\cup} \ldots \dot{\cup} C_M = V$, Class labels of each cluster $x : \{C_i\} \to [K]$,

1 Initialize clustering: $C_i = \{i\} \, \forall i \in V$
2 Initialize class labels: $x(C_i) = \arg\min_{k \in [K]} c_V(i, k) \, \forall i \in V$
3 **while** $E \neq \varnothing$ **do**
4      Best merge candidate: $ij = \arg\max_{ab \in E} \texttt{total\_edge\_similarity}(ab)$
5      **if** $t_{ij} < 0$ **then**
6          **break**
7      Merge $j$ with $i$: $C_i = C_i \cup C_j, C_j = \varnothing$
8      Remove edge $ij$: $E = E \setminus \{ij\}$
9      Assign joint class label: $x(C_i) = k_{ij}$
10      Update node costs: $c_V(i, k) = c_V(i, k) + c_V(j, k) \, \forall k \in [K]$
11      Assign neighbours of $j$ to $i$:
12      **for** $jh \in E$ **do**
13          **if** $ih \notin E$ **then**
14              $E = E \cup ih$
15              $c_E(ih) = 0$
16          $c_E(ih) = c_E(ih) + c_E(jh)$
17      **end**
18 **end**
19 Merge non-partitionable clusters:
20 **for** $C_i, C_j : x(C_i) = x(C_j), x(C_i) \notin P$ **do**
21      $C_i = C_i \cup C_j$
22      $C_j = \varnothing$
23 **end**
24 **return** $\{C_i\}_i, x$
25 **Function** $\texttt{total\_edge\_similarity}(ij)$
26      Best joint class label: $k_{ij} = \arg\min_{k \in [K]}[c_V(i, k) + c_V(j, k)]$
27      Merge cost: $s(ij) = c_V(i, k_{ij}) + c_V(j, k_{ij})$
28      Separation cost: $m(ij) = c_E(ij) + c_V(i, x(C_i)) + c_V(j, x(C_j))$
29      Compute similarity: $t(ij) = m(ij) - s(ij)$
30      **return** $t(ij)$

---

## A.2  Training details

Table 2 contains the hyperparameters used for fully differentiable training. The data augmentation scheme is the same as the one used in Panoptic-DeepLab in [70] (which we also use during baseline training). Since the panoptic quality surrogate loss is in $[0, 1]$ we multiply it by a scalar $w$ which in turn affects the magnitude of perturbation in the costs (Line 1 in Alg. 1). As the COCO dataset contains significantly more classes than Cityscapes, we scale the loss by a larger number to ensure that its magnitude is large enough. Notice that the gradient estimates (Lines 4-5 in Alg. 1) would always be in $[-1, 1]$ irrespective of loss scaling.

Table 2: Hyperparameters for fully differentiable training. $R(a, b, c)$: all values in $[a, b]$ divisible by $c$.

| Dataset | Optimization | | | | | Data aug. | | |
|---|---|---|---|---|---|---|---|---|
| | $\lambda$ | $N$ | $w$ | LR | $D\%$ | Crop | Horiz. flip | Resize |
| Cityscapes | [1, 5e3] | 5 | 10 | 1e-3 | 10 | 512, 1024 | ✓ | R(512, 2048, 32) |
| COCO | [1e3, 5e3] | 5 | 100 | 1e-4 | 10 | 640, 640 | ✓ | R(448, 768, 64) |

Table 3: Statistics of our models for Cityscapes and COCO datasets. We also report runtime breakdown (in seconds) for one training iteration with batch size 24. Overall time includes both forward and backward pass.

| Dataset | Params | Time for 1 training itr. | | |
|---|---|---|---|---|
| | | AMWC | MWC | Overall |
| Cityscapes | 34M | 2.5 | 6.1 | 16.8 |
| COCO | 34M | 3.5 | 9 | 15.7 |

Lastly, we randomly set $10\%$ of values in $c_V, c_E$ to zero (indicated by $D$ in table 2) during fully differentiable training which makes learning harder. This is similar to dropout except that it is applied to the costs of an optimization layer and secondly the costs are not normalized by dropout rate during test time. This gives a slight but consistent improvement of about $0.3$ points in PQ ($\%$) during evaluation.

Model statistics are shown in Table 3 showing number of trainable parameters and runtime for one training iteration where we compare time spent on solving AMWC and MWC problems during forward and backward pass resp.

## A.3 Loss on AMWC

Here we perform an ablation study where we directly apply loss on semantic class labels $x$ and edge labels $y$ instead of panoptic labels. Since we do not use panoptic labels, this approach does not require transformation to MWC. The gradients can be computed by perturbing associated semantic costs $c_V$ and edge costs $c_E$ and calling the same AMWC solver in the backward pass. Given ground-truth labels $x_g, y_g$, the losses are

$$L_V = \frac{1}{|V|}\|x - x_g\|_1 \tag{8}$$

$$L_E = 1 - \frac{y^T y_g}{y^T y_g + 0.5(y^T(1 - y_g) + (1 - y)^T y_g)} \tag{9}$$

Here the loss on edge labels is based on the F1-score following the approach of SMW [69] to account for class-imbalance. The loss (9) is applied separately on each affinity classifier. Afterwards the approach of [64] can be directly applied to compute gradients except that we use $N = 5$ using the robust backpropagation formula (4) for a fair comparison with the panoptic quality surrogate. Lastly, the losses are scaled to put more emphasis on small objects and 'thing' classes in the same way as done for baseline pre-training.

We conduct a comparison on Cityscapes dataset and train using the same setup as for the panoptic quality surrogate loss and use the checkpoint with lowest validation error. Results are given in Table 4. We can see that optimizing PQ surrogate gives better performance and using separate losses decreases the performance especially on 'thing' classes. This is due to multiple reasons: (a) The loss applied on affinities cannot perform well w.r.t. PQ because each edge mis-classification is penalized arbitrarily instead of calculating its impact on PQ, (b) a slight localization error in boundary detection is penalized in the same way as boundary localization errors.

Table 4: Comparison of PQ surrogate loss with separate losses on AMWC output

| Loss | PQ | PQ$_{th}$ | PQ$_{st}$ |
|---|---|---|---|
| Separate losses | 57.8 | 45.7 | 66.6 |
| PQ surrogate | 62.1 | 55.1 | 67.2 |

## A.4   Reproducibility

To ensure that results of fully differentiable training are reproducible we finetune our baseline with 6 random seeds on the Cityscapes dataset for 1500 iterations (instead of 3000 for our main results) and evaluate on the validation split. This introduces multiple sources of randomness in the training process due to mini-batch selection, drop-out, data augmentation etc. More importantly the values of $\lambda$ in (4) change since they are also drawn randomly in an interval. The results are contained in Table 5 showing that all trials improve over the baseline by fully differentiable training.

Table 5: Reproducibility of fully differentiable training after 1500 iterations under random seeds on Cityscapes validation set. For comparison we also show performance of baseline and fully differentiable training after 3000 iterations.

| Trial | PQ | PQ$_{th}$ | PQ$_{st}$ |
|---|---|---|---|
| 1 | 61.41 | 54.40 | 66.50 |
| 2 | 62.01 | 54.78 | 67.26 |
| 3 | 61.77 | 54.73 | 66.89 |
| 4 | 61.33 | 54.37 | 66.40 |
| 5 | 61.91 | 54.85 | 67.06 |
| 6 | 62.07 | 55.14 | 67.11 |
| Baseline | 58.5 | 48.3 | 66.0 |
| Fully differentiable (final) | 62.1 | 55.1 | 67.2 |

## A.5   Affinity classifiers

The affinity feature maps $f_A$ from the affinity decoder and node costs $c_V$ from the semantic segmentation branch are used for computing affinity scores. First $f_A, c_V$ are concatenated and reduced to 256 channels by two convolutional layers. Afterwards, the result is sent to each classifier specific to an edge distance $d$. Each classifier predicts horizontal and vertical edge affinities. These steps are illustrated in Figure 7.

For long-range edges, we first take the difference of node features. Specifically, given node features $f$ of shape $B \times N \times H \times W \to \mathbb{R}$ (where $B, N, H, W$ correspond to batch-size, channels, height, width resp.), horizontal and vertical edge features $g_h^d, g_v^d$ for a distance $d$ are computed as

$$g_h^d(b, n, i, j) = f(b, n, i, j + \lfloor \frac{d}{2} \rfloor) - f(b, n, i, j - \lceil \frac{d}{2} \rceil) \tag{10}$$

$$g_v^d(b, n, i, j) = f(b, n, i + \lfloor \frac{d}{2} \rfloor, j) - f(b, n, i - \lceil \frac{d}{2} \rceil, j) \tag{11}$$

This operation is marked by $\mathcal{S}_d$ in Figure 7. Afterwards, we make use of depth-wise separable convolution with 2 groups for efficiency. Note that the indexing in (10) is done in such a way that center locations of each horizontal and vertical edge match, see Figure 8. The reason is that if there is an oblique boundary in an image there is a high chance that both horizontal and vertical affinities would be low. To capture this inter-dependence the last layer of each affinity classifier does not use depth-wise separable convolution.

## A.6   Other evaluation metrics

Table 6 contains results of instance segmentation and semantic segmentation evaluation metrics on the Cityscapes dataset. On the instance segmentation task our fully differentiable approach performs

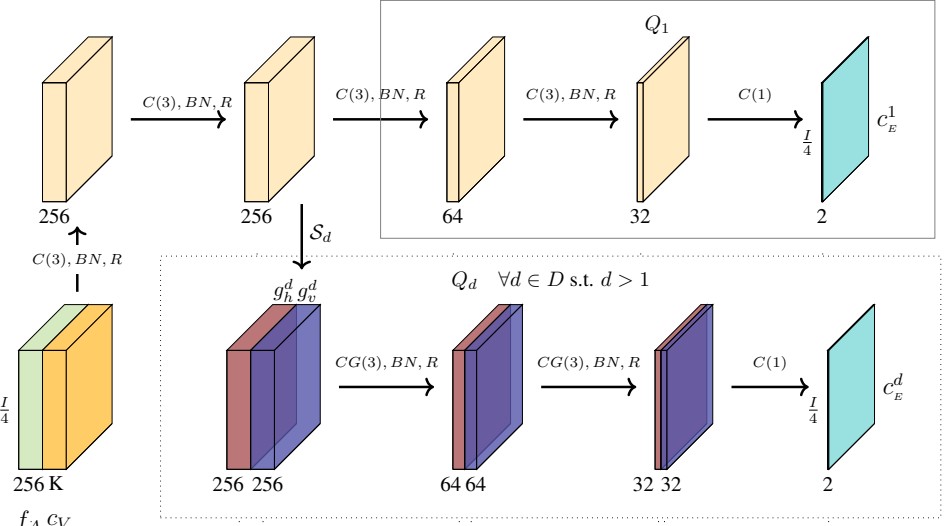

Figure 7: Classifier $Q_1$ predicts information about the finest scale (similar to edge detection in images). In-addition there are $|D-1|$-many classifiers (one is shown in dotted region) for long-range context. All classifiers produce a two channel output $c_E^d$ containing horizontal and vertical edge costs at a distance $d$. $\mathcal{S}_d$: takes differences of node features in $+$ neighbourhood with edge distance $d$ giving $g_h^d, g_v^d$. ($C(n)$: $n \times n$ conv., $BN$: batch-norm, $R$: ReLU, $CG(n)$: $C(n)$ with 2 groups.)

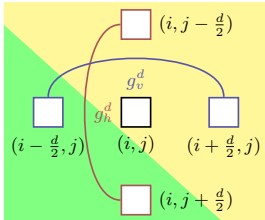

Figure 8: Edge neighbourhood around a location $i,j$ where the image contains two regions indicated by yellow and green colors. Assuming edge distance $d$ is even. Horizontal and vertical features $g_h^d, g_v^d$ at location $i,j$ are computed from edge end-points. Note that both horizontal and vertical edges would have low affinity values due to presence of a boundary.

better than all approaches which use ResNet-50. For the semantic segmentation task we also get a slight improvement over the baseline by fully differentiable training.

### A.6.1    Instance segmentation evaluation

Average precision (AP) is used to assess instance segmentation performance. To calculate AP one additionally requires uncertainty scores for each instance to establish a ranking. We compute the uncertainty score for an instance with mask $p \in \{0,1\}^{|V|}$ having class label $l$ as

$$\frac{1}{|P|}\sum_{i \in V} c_V(i,l)p(i) + \underbrace{\frac{\sum_{ij \in E} 1[p(i) \neq p(j)]c_E(ij)}{\sum_{ij \in E} 1[p(i) \neq p(j)]}}_{\text{Inter-cluster mean similarity}} - \underbrace{\frac{\sum_{ij \in E} 1[p(i) = p(j)]c_E(ij)}{\sum_{ij \in E} 1[p(i) = p(j)]}}_{\text{Intra-cluster mean similarity}} \tag{12}$$

### A.6.2    Comparison w.r.t boundary-based quality metrics

We additionally evaluate the performance of our model using the metrics proposed in [17] which focuses more on the quality of detected boundaries. The results are given in Table 7 which shows the despite downsampling our fully trained model can still outperform methods such as Panoptic-

Table 6: Instance and semantic segmentation performance (AP, mIoU resp.) on Cityscapes validation set. †: Mask selection (e.g. by Mask-RCNN), *: Uses test-time augmentation. (-) Marks the results which are not reported for that setting.

| Method | Backbone | AP(%) | mIoU(%) |
|---|---|---|---|
| EfficientPS [52]† | Custom | 38.3 | 79.3 |
| Panoptic-FPN [36]† | ResNet-101 | 33 | 75.7 |
| UPSNet [73]† | ResNet-50 | 33.3 | 75.2 |
| Unify. PS [46]† | ResNet-50 | 33.7 | 79.5 |
| Panoptic-DL [16] | Xception-71 | 35.3 | 80.5 |
| Axial-DL [66] | Axial-L | 35.8 | 81.0 |
| Panoptic-DL [16] | ResNet-50 | 33.1 | 78.1 |
| SSAP [27]* | ResNet-101 | 37.3 | - |
| SSAP [27] | ResNet-50 | 31.5 | - |
| COPS baseline | ResNet-50 | 32.7 | 78.5 |
| COPS fully differentiable | ResNet-50 | 34.1 | 79.3 |

DeepLab [16] in terms of boundary quality. We can also see that most of the performance gain of full training over baseline actually comes from increased recognition quality (RQ).

Table 7: Evaluation w.r.t boundary based panoptic quality metrics [17] denoted by subscript 'b' computed on Cityscapes validation set

| Method | $PQ_b$ | $SQ_b$ | $RQ_b$ | PQ | SQ | RQ |
|---|---|---|---|---|---|---|
| Panoptic-DL [16] | 36.3 | **64.3** | 55.6 | 60.3 | 81.5 | 72.9 |
| COPS baseline | 35.2 | 62.3 | 55.9 | 58.5 | 80.3 | 71.8 |
| COPS full | **38.9** | **64.3** | **59.7** | 62.1 | 81.5 | 75.2 |

## A.7 AMWC without downsampling

Due to runtime issues associated with AMWC solver our results are computed at $1/4$-th of the input resolution. To quantify the potential gains in quality we report results computed without this downsampling. Results are given in Table 8 where we see improvement in all metrics although with a significant slow down due to sequential nature of AMWC solver. Nonetheless, this shows that our results can be further improved if faster algorithms for solving AMWC are developed.

Table 8: Effects of downsampling in COPS evaluation performance computed on Cityscapes validation set. The runtime is computed for batch-size of 1.

| Downsampling factor | PQ | SQ | RQ | AP | Runtime(sec.) |
|---|---|---|---|---|---|
| 1/4 | 62.1 | 81.5 | 75.2 | 34.1 | 2 |
| 1 | 63.1 | 82.9 | 75.4 | 38.2 | 15 |

## A.8 Oracle study

We perform an oracle study where the segmentation costs $c_V$ sent as input to AMWC are replaced by ground-truth. This helps in establishing an upper bound on performance assuming that the semantic segmentation branch is performing perfectly. The affinity costs $c_E$ are still computed through the network. Results are given in Table 9.

We see a substantial increase in PQ scores for COCO dataset showing that panoptic segmentation performance on COCO dataset is heavily influenced by semantic segmentation as it contains a large number of classes (133). Since the affinity costs can only make cut/merge decisions for a pair

of pixels, it cannot be a major source of improvement in semantic performance (except in edge localization).

Lastly, we do not see $100\%$ score in $\mathrm{PQ^o_{st}}$ because our results are downsampled by a factor of $1/4$ w.r.t. the ground-truth.

Table 9: Oracle study: Evaluation on subset of validation split of COCO and Cityscapes.

| Dataset | Actual results | | | Semantic Oracle | | |
|---|---|---|---|---|---|---|
| | PQ | $\mathrm{PQ_{th}}$ | $\mathrm{PQ_{st}}$ | $\mathrm{PQ^o}$ | $\mathrm{PQ^o_{th}}$ | $\mathrm{PQ^o_{st}}$ |
| Cityscapes | 62.1 | 55.1 | 67.2 | 81.1 | 67 | 91.4 |
| COCO | 38.4 | 40.5 | 35.2 | 70.9 | 57.2 | 91.2 |

## A.9 Design choices for panoptic segmentation

We argue that approaches should be compared not only in terms of their performance on benchmarks but also w.r.t. other factors such as network complexity, number of hyperparameters etc., which also matters in production. In table 10 we compare different panoptic segmentation approaches in terms of these properties. Moreover, we also mention whether these approaches can be trained end-to-end (which reduces the number of hyperparameters during training) and whether they optimize the metric-of-interest (i.e., panoptic quality). Even though for a real-world application one might not want to optimize panoptic quality directly, it can serve as a starting point for devising a metric one cares about in production.

Table 10: Qualitative comparison of different approaches of panoptic segmentation in terms of neural network (NN) complexity, number of hyperparameters, end-to-end differentiability and whether they optimize panoptic quality at training time. Last column indicates performance on validation sets of corresponding datasets.

| Methods | Complexity | # of Hyperparams. | | E-to-E | Opt. PQ | PQ | |
|---|---|---|---|---|---|---|---|
| | | Train | Eval | | | Citysc. | COCO |
| Max-DL [65] | High | Less | Less | ✓ | Partially | - | 49.3 |
| Eff. PS [52] | High | Many | Many | ✗ | ✗ | 63.9 | - |
| UPSNet [73] | High | Many | Less | ✗ | ✗ | 59.3 | 42.5 |
| Unify. PS [46] | High | Less | Less | ✓ | Partially | 61.4 | 43.4 |
| Axial-DL [66] | Medium | Less | Less | ✗ | ✗ | 63.9 | 41.8 |
| SSAP [27] | Medium | Less | Less | ✗ | ✗ | 61.1 | 36.5 |
| SMW [69] | Medium | Many | Less | ✗ | ✗ | 59.3 | - |
| Panoptic-DL [16] | Low | Low | Less | ✗ | ✗ | 60.2 | 35.1 |
| COPS baseline | Low | Low | None | ✗ | ✗ | 58.5 | 34.3 |
| COPS full | Low | Low | None | ✓ | ✓ | 62.1 | 38.4 |

## A.10 Example results

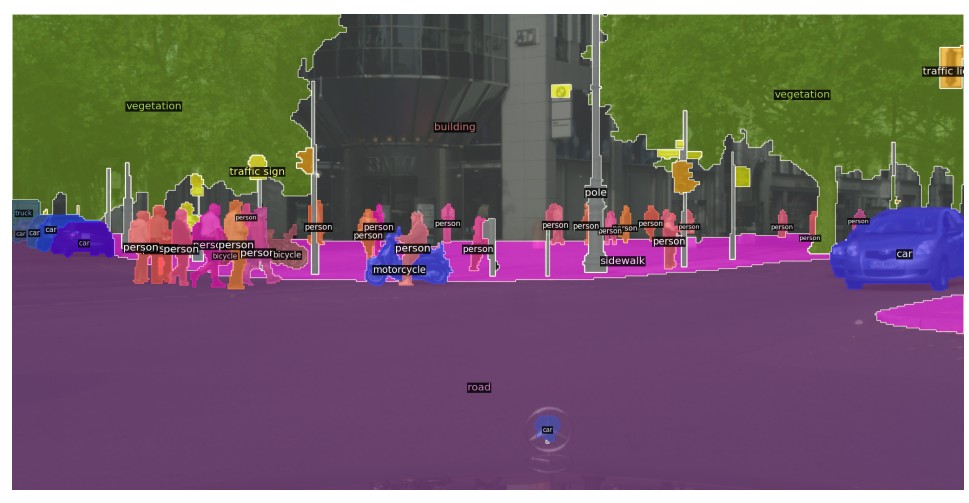

(a)

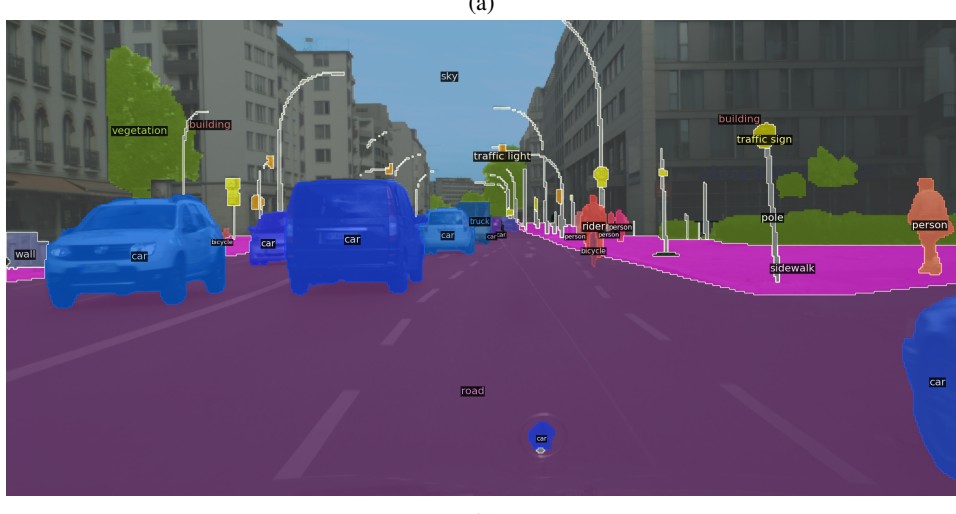

(b)

Figure 9: Example results on Cityscapes test set