# OpenReview forum: "Combinatorial Optimization for Panoptic Segmentation: A Fully Differentiable Approach"
_NeurIPS.cc/2021/Conference — NeurIPS 2021 Poster_

### Official Review · Reviewer_8yaS · 2021-07-05

**Rating:** 5
**Confidence:** 4

**Summary:**

This paper introduces to use combinatorial optimization method to solve the panoptic segmentation problem. The perturbation technique is used for training with the Asymmetric Multiway Cut (AMWC) solver. Furthermore, a surrogate panoptic loss is introduced and works together with the proposed framework. Experiments are performed on public COCO and Cityscapes panoptic segmentation datasets.


**Limitations And Societal Impact:**

Limitations are discussed to some extent, such as the inference time of the proposed method. The method requires heavy hyper-parameter tuning from my view, which could also be a limitation.

**Main Review:**

Pros:

1. Given the challenge of panoptic segmentation, using the combinatorial optimization method for the panoptic segmentation is novel, although this kind of method has been explored earlier in Instance Segmentation.

2. The proposed panoptic loss is meaningful since it optimizes the model directly towards the evaluation metric.

Cons:

1. The presentation of this paper is poor. Readers may have a hard time following the proposed method.

    a. The meaning of the symbols is not clearly explained and confusing. For instance, there are no explanations of $l$, $i$, $j$, $C$ in Equation (1), $x$ in Equation (2), $L$ in Line-173 (while in Line-193, $L$ is used again for representing a loss function). Readers would have to guess the meaning of the symbols or refer to the referenced papers so as to understand the equations. The statements "$\mathcal{X}$ is some constraint set", "$\lambda$ are sampled uniformly on some interval. " are vague and unclear.

    Besides, in Line-219 to Line 220, the funtions $h(u) = \frac{u^4}{u^4 + (1 - u)^4}$ and function $\sigma_l(p) = [1 + e^{-0.1(1^{T}p - t_l)}]^{-1}$ seem to be very ad-hoc. How are they derived?

    b. Section 2.2 only categorized a few related works. How does this work connect with each category of the mentioned works?

    c. It is said in the paper: "We cross-validate our method on the validation split and evaluate the same model on the testing split". However, in Table 1, validation results are also used to compare with other methods, which may cause confusion.

2. The claims of this paper are not well supported in the experiments. There are nearly no detailed ablations over each part of the proposed method. It is hard to tell why the proposed method works since there are only comparisons with other methods and the baseline. For instance, does the proposed Combinatorial Optimization solver need to be tightly coupled with the panoptic surrogate loss? If not, how does it perform compared with baseline when training without the panoptic surrogate loss? How would it perform when using the panoptic surrogate loss on Panoptic-Deeplab?

Furthermore, the paper claims it is an end-to-end training approach. However, from my point of view, it is not end-to-end, but using stage-wise training. It needs to pretrain the baseline and then fix the network to train the solver.

3. Besides, I have further questions about some unclear details:

    a. In Figure 4, it only shows the comparison based on $\bar{PQ}$ rather than $PQ$, why? Would $\bar{PQ}$ perfectly track the $PQ$ evaluation?

    b. Why only train 4000 iterations on COCO in the so-called "end-to-end training" phase? It seems too short (less than 1 epoch). Would more training lead to worse results?



**Time Spent Reviewing:**

8

---

> ### Author Response · Authors · 2021-08-08
> **Rebuttal**
>
> We thank you for the detailed review and pointing out potential issues.
>
> * **Writing:** We will improve the overall writing and make it more precise. Thank you for pointing out the problematic passages.
>
> * **Typos, inconsistencies:** Thanks for pointing out the errors. We will address the raised points.
>
> * **Ad-hoc functions (line 219,220):** We agree that other approximations to the panoptic quality are certainly possible. However, the one we present is, given the involved nature of the panoptic quality metric, still rather simple and nicely interpolates it.
>
> * **Comparison with related work:** In comparison to related work we strove to have a simpler method that elegantly incorporates the final segmentation step in the end-to-end training and results in corresponding performance gains. In particular, our method has fewer hyperparameters and is simpler than most of other panoptic segmentation approaches e.g. UPSNet [67] uses a Mask-RCNN and additionally has eight loss terms which need to be balanced. Please see Table 3 in the supplementary material for a thorough comparison.
>
> * **Cross-validation:** We track the panoptic quality surrogate on ten percent of the validation split during training for early stopping. This reduces the chance of overfitting on the validation set. Nonetheless, the ablation study done in Table 5 in the supplementary material and our results on testing splits provide further evidence for robustness. We will clarify this point to resolve any potential confusion in the final manuscript.
>
> * **The claims of this paper are not well supported, no ablations:** We have three main contributions for which we also provide ablations:
>     1. _End-to-end training:_ Our baseline is trained through conventional way similar to [23, 63]. Moreover we use different loss function for pretraining which already allows us to exceed the performance of [23, 63] on comparable backbones. This proves that our baseline is already very competitive. Since end-to-end training (without conventional losses) even surpasses this baseline it indicates that the performance gains are made by our main contributions without any confounding factor.
>     2. _Panoptic quality surrogate loss:_ The ablation study which does not require this loss (and conversion to MWC) inspired from [63] can be found in Table 4 of the appendix.
>     3. _Robust backpropagation:_ Figure 4 shows that our improvement to [52] leads to faster convergence.
>
> * **Combinatorial Optimization solver needs to be coupled with the panoptic surrogate loss:** We are not aware of any work for panoptic segmentation which produces hard panoptic labels in a differentiable manner. For example, PanopticDeepLab[13] uses a voting scheme for post-processing which is not differentiable (as also mentioned in [13] pg. 8). Therefore, if a differentiable method is devised in the future which is different than ours then our loss function can be used directly and tested independently of our optimization layer.
>
> * **Stage-wise training:** Please refer to the corresponding point in the overall comments section.
>
> * **Training loss in Figure 4:** The primary reason of the plot is to show that our robust pertubation technique converges faster and more importantly the approximated gradients can minimize the training loss. Empirically, we did observe panoptic quality surrogate closely following the panoptic quality metric which we will additionally include in Figure 4. Please also see answer to _upv8_ for the related question.
>
> * **4000 iterations on COCO in the so-called "end-to-end training":** Training more on COCO dataset does not decrease the training loss further. We believe that it is because the performance on COCO dataset is primarily tied to semantic segmentation (due to a large number of classes) for which our strong baseline using loss from [11] is already effective. Further improvements require changes in the neural network architecture. Please also see the oracle study in section A.8.

---

> > ### Comment · Reviewer_8yaS · 2021-08-21
> > **Responce**
> >
> > Thank you for your reply. I have read the authors' rebuttal and other reviewers' comments. The rebuttal partially resolves my concerns about the method part. The maximum score I would like to rate is 5.
> >
> > However, I still have the following concerns that make me believe the paper is below the line:
> >
> > 1). Overclaim, as also pointed out by reviewer aCGb.
> >
> > - The authors insist that the method is "end-to-end training". However, I still think the title and the main contribution, "An End-to-End Trainable Approach", may cause confusion.  Since the pre-training phase is also part of the method and it is indispensable, the overall approach is not "end-to-end trainable" but a "pre-training then end-to-end fine-tuning" approach.
> >
> > - The paper claims the method is simpler because it requires fewer hyper-parameters. From my point of view, it requires a pre-training phase, which uses a similar training strategy as PanopticDeeplab. And the second phase also requires additional hyper-parameters, such as the smoothing function in Line 219-220. It provides a new approach, but it is not simpler than previous works.
> >
> > 3). Changes promised by the authors may need another round of review, including explanations of equations and the baseline, adjustment of results (*e.g.*, removing all improper comparisons on validation set), adjustment of Figure 4.

---

> > > ### Author Response · Authors · 2021-08-24
> > > **Training logs and end-to-end training**
> > >
> > > Thank you for considering our points. We would like to provide more information about the following:
> > >
> > > **Experiment logs:** We have exported our experiment logs at [tensorboard.dev](https://tensorboard.dev/experiment/8P37I2HRTXOtrK9UeMilhg/). We will also add these plots in the paper. We would like to highlight following points:
> > > 1.  **PQ v/s PQ surrogate**:  Training metrics (_prefix:_ train) i.e., $PQ$ and our surrogate $PQ_{surr}$ are both increasing. Note that the scale of values are different because $PQ$ uses a greedy matching strategy with hard thresholding while $PQ_{surr}$ uses bipartite matching on our smoothed matching scores.
> > > 2.  **Validation set**: We additionally show results on the Cityscapes validation set (containing 500 images) during training (_prefix:_ test). We can see that $PQ$ is increasing in a smooth fashion showing that careful model selection is not necessary.
> > >
> > > **End-to-end training:** We understand the reviewer's concerns regarding the usage of this term and we acknowledge that there might be misunderstandings.
> > > 1. It has been reported by [1] that training of all parameters has its limitations where in the worst case nothing is learned. The author suggests an alternative procedure where simpler modules are learned first and their weights are frozen for subsequent learning stages. This is the strategy we also follow.
> > > 2. We believe that the term end-to-end training is used loosely in the literature. To name two examples, the architecture in [2] is called end-to-end trainable even though only some parameters of the are trained. The work [3] similarly to ours first pre-trains and then fine-tunes through the combinatorial solver (but does not freeze the backbone as we do).
> > >
> > >     To better clarify the scope of our work we have thought about using the term _fully differentiable_ instead of _end-to-end_, similarly as in [4]. If there is consensus among reviewers that the term fully differentiable is more appropriate, we would be happy to change our manuscript accordingly.
> > >
> > > [1] Limits of End-to-End Learning. Tobias Glasmachers. ACML 2017.
> > >
> > > [2] Hierarchical Aggregation for 3D Instance Segmentation. Shaoyu Chen, Jiemin Fang, Qian Zhang, Wenyu Liu, Xinggang Wang, ICCV 2021
> > >
> > > [3] Deep Graph Matching via Blackbox Differentiation of Combinatorial Solvers. Michal Rolinek, Paul Swoboda, Dominik Zietlow, Anselm Paulus, Vit Musil, Georg Martius. ECCV 2020
> > >
> > > [4] Learning a Neural Solver for Multiple Object Tracking. Guillem Braso, Laura Leal-Taixe. CVPR 2020.

---

### Official Review · Reviewer_G2zw · 2021-07-06

**Rating:** 7
**Confidence:** 5

**Summary:**

This paper formulates panoptic segmentation as a combinatorial optimization problem that uses an asymmetric multiway cut problem (AMWC) solver. The authors propose a bottom-up approach that builds segmentations by combining CNNs and AMWC which is inspired by the idea of combing optimization and neural network. The model is trained with surrogate loss that optimizes directly the final panoptic quality metric, making the model end-to-end trainable.

**Limitations And Societal Impact:**

yes

**Main Review:**

This paper brings up an interesting idea of formulating panoptic segmentation as multiway cut problem and embed the optimization problem into the network and results show the effectiveness of the model. I only have a few concerns/questions to the authors:

1. According to L318-320, the AMWC solver requires 2 seconds per image, making the proposed method not very practically useful. This is probably the biggest limitation of the proposed approach (other models with extremely large backbone probably don't need 2 seconds to process an image). What is the benefit of using AMWC compared to existing methods like [13]? Is there anything AMWC can achieve but [13] cannot achieve? I suggest the author to make a more detailed error mode analysis between their methods and [13].

2. Boundary quality issue. From visualization in Figure 5, the boundary quality is not very good. I understand this is due to the nearest neighbor upsampling on discrete grid. Is it possible to solve AMWC after upsampling the intermediate features (like semantic segmentation and affinity prediction)? I don't see any discussion on the mask quality other than L273. PQ improvement comes from both recognition (RQ) and segmentation (SQ) qualities and according to Figure 5 the proposed method mainly improves recognition quality? I suggest also do a thorough analysis of mask quality with [13] as the mask predictions in [13] seems to be smoother. Recently, Boundary PQ is proposed in [A] to measure the boundary quality which could be a useful metric. I think it would be good to point out the possible limitation by comparing Boundary PQ.

3. According to 4.2, the end-to-end training is actually finetuning from some pre-trained weights. Is it possible to remove the necessity of this two-stage training?

[A] Cheng, Bowen, et al. "Boundary IoU: Improving Object-Centric Image Segmentation Evaluation." Proceedings of the IEEE/CVF Conference on Computer Vision and Pattern Recognition. 2021.

**Time Spent Reviewing:**

2 hours

---

> ### Author Response · Authors · 2021-08-08
> **Rebuttal**
>
> We thank you for a detailed review.
> * **Inference time:** See the corresponding point in the overall comments section.
>
> * **Benefits of AMWC:** Other panoptic segmentation approaches require parameters which cannot be trained and thus they cannot be trained end-to-end. For example:
>     1. PanopticDeepLab [13] needs non-maximum suppression, probability thresholding which introduces additional hyperparameters. Our pipeline does not have such non-trainable parameters which allows us to exceed the performance of [13]. Moreover the voting scheme of [13] can suffer when two objects have the same center location and one is occluded by another.
>     2. The work [23] uses a coarse-to-fine scheme and fuses results from eachs scale together, while we only make one call to our AMWC solver.
>
>     Our pipeline is also easier to adapt to different scenarios as it has less hyperparameters. Lastly, we argue that our approach is more theoretically appealing than ad-hoc methods such as voting schemes.
>
> * **Solving AMWC at finest resolution:** This is a good idea. However, solving AMWC at finest resolution is computationally challenging due to two reasons. Firstly, AMWC takes around $15$ seconds per image on finest resolution. Second, upsampled affinity features need to be further processed by the affinity classifier which makes inference slow and requires more GPU memory. Nonetheless we tried this and got the following results on Cityscapes validation set:
>
>   | Resolution | PQ | SQ | RQ | AP |
>   | --- | -- | -- | -- | -- |
>   | Coarse | 62.2 |  81.4 | 75.4 | 35   |
>   | Finest | 63.1 | 82.9 | 75.4 | 38.2 |
>
>   This shows that performance can be improved further through upsampling although this comes at the cost of a significant computational burden. Perhaps more intelligent upsampling mechanisms can be employed to address this issue.
>
> * **Improvement in RQ/SQ and boundary PQ:** Thanks for pointing this out and for the reference to [A]. The reviewer is correct in stating that most of the improvements in PQ are due to recognition quality (RQ) as compared to semantic quality (SQ) (which also improves). Since we are optimizing the panoptic quality surrogate directly the loss function can decide which part of the quality metric can be optimized further during training. The results on Cityscapes validation set are shown below (**_b** stands for boundary based quality metrics computed from [A]).
>
>   | Method       | PQ | SQ | RQ | PQ_b | SQ_b | RQ_b |
>   | ---------- | -- | -- | -- | -- | -- | -- |
>   | [13]       |  60.3  | **81.5**   | 72.9   |  36.3 | **64.3** | 55.6 |
>   | Our baseline   | 58.5 | 80.3 | 71.8 | 35.2 | 62.3 | 55.9 |
>   | Our end-to-end | **62.2** | 81.4 | **75.4** | **38.9** | **64.3** | **59.7** |
>
>
> * **Two-stage training:** Please see the corresponding point in the overall comment section.

---

> > ### Comment · Reviewer_G2zw · 2021-08-22
> > **Response**
> >
> > I thank the authors for their detailed response. The extra analysis addresses my concern, and I recommend the authors consider putting them in the paper (or supplementary material). Although this paper has some limitations as pointed out by other reviewers, I still believe this paper has value to the community and I'm leaning towards accepting this paper and I will raise my score to 7.

---

### Official Review · Reviewer_upv8 · 2021-07-15

**Rating:** 7
**Confidence:** 4

**Summary:**

This paper proposes an approximated (smoothed) end-to-end training method for panoptic segmentation with combinatorial optimization. Panoptic segmentation is obtained by AMWC, and the gradients are estimated with robust perturbation. Positive experimental results are achieved on two panoptic segmentation datasets, compared with the baselines.


**Limitations And Societal Impact:**

Yes.

**Main Review:**

## Originality
To the best of my knowledge, the proposed AMWC approach for panoptic segmentation is original and novel.

## Strength
1. The neural network architecture is simple, with a semantic head and an intuitive affinity head.
2. Overall presentation and experiment settings are clear, although a few details and ablation experiments are in the appendix.
3. Optimizing directly on panoptic quality does not seem to overfit PQ. The method works well on other metrics (mask AP, mIoU).
4. The experiments show a comfortable margin on top of the baselines. And the control experiments are fair.

## Weakness
1. The inference speed is bottlenecked by AMWC solver, limiting the application of the proposed method in real world. The authors claim the training time AMWC/MWC is computed in parallel, but it is not clear how much more time is spent on them.
2. The proposed method may be limited to hard assignment problems only. It may not generalize to more flexible cases such as COCO instance segmentation with potentially overlapping output masks.
3. Baselines are relatively weak, with resent-50 and panoptic-deeplab. It is not clear if the end-to-end training still leads to a good margin on top of other strong methods, e.g. max-deeplab.
4. Two stage training is required, possibly because of the slow AMWC/MWC CPU solver which limits the scaling to more GPUs training in parallel.
5. Training time approximated PQ might still be very different from real PQ probably because of the different matching strategies (bipartite vs. IoU thresholding). This might make the validation PQ a less good indicator of the real PQ.

———————————————— Post Rebuttal ——————————————

I thank the authors for the responses. The rebuttal partially resolves my concern (weakness 5, and part of weakness 1). The authors and most of the reviewers seem to agree on other limitations (e.g. the limited efficiency and thus the two-stage training problem). I think these limitations are acceptable at the first attempt, so I will raise my rating to 7 (accept).

**Time Spent Reviewing:**

3

---

> ### Author Response · Authors · 2021-08-08
> **Rebuttal**
>
> We thank you for the positive appraisal, insightful review and criticism of our paper.
>
> * **Training time breakdown:** The training time breakdown per batch (for batch size of 24) is as follows:
>
>     | Stage | time (s) |
>     | --- | -: |
>     | AMWC (forward) | 3 |
>     | MWC (backward) | 8 |
>     | Overall | 16 |
>
>     Note that we do not parallelize over $N = 5$ in the backward pass for MWC solver. The overall time also includes time for loss calculation and bipartite matching.
>
> * **Hard assignment & overlapping output masks:** In the panoptic segmentation problem the metric accepts exactly one label per pixel, so having multiple assignments is not allowed in this case. We agree however that in other related segmentation scenarios multiple labels per pixel can be desirable. We believe our approach can be extended to this setting: In this case a variation of the AMWC problem without label exclusion constraints would need to be employed with the loss function of interest. We believe this would be an interesting extension of our work.
>
> * **Baselines are weak:** We argue that our baseline is not weak. It outperforms [13, 63] on comparable backbones.
>
> * **Comparison with strong methods:** The goal of our study was not to get an overall best performance by employing state-of-the-art neural networks (which MaxDeepLab [59] does by using transformers). We wanted to tackle the challenging task of combining an optimization module with a neural network by enabling end-to-end training. Our study shows a way to achieve such goal and its benefits. Since the bottom-up approaches [13, 23, 64] most relevant for our comparison used ResNet-50, we believe that to be the common denominator that would best showcase our contribution in comparison to competing methods. One can exchange neural network blocks to improve the performance, yet retain end-to-end training and its benefits. Such larger experimental study was however beyond the scope of our work.
>
> * **Two-stage training:** See the corresponding point in the overall comments section.
>
> * **Training time PQ surrogate fidelity, bipartite matching vs IoU matching:** The IoU matching introduces a hard thresholding and therefore is non-differentiable. Therefore, we had to introduce a surrogate matching process that is differentiable. However, our bipartite matching scheme approximates IoU thresholding well: If all the instances have IoU > 0.5 to ground truth segmentations, bipartite matching and IoU matching give identical assignments. Empirically, we also show that the original PQ metric improves after training w.r.t. the surrogate PQ (see Table 1), giving evidence that our approximation, including bipartite matching, is faithful.

---

### Official Review · Reviewer_UruC · 2021-07-16

**Rating:** 6
**Confidence:** 3

**Summary:**

This paper proposes a new model architecture for panoptic segmentation which is end-to-end trainable and requires less hyperparameters than existing methods.

**Limitations And Societal Impact:**

Yes, the authors clearly show the results of their method and the compute time necessary to train and run inference.

**Main Review:**

This paper proposes a new model architecture for panoptic segmentation.  The proposed architecture produces semantic segmentations and edge affinity scores from every pixel to its neighboring pixels. The edge affinities help determine whether two pixels belong to the same instance. The segmentations along with the edges allow us to view the image as a graph (V, E) where the vertices are pixel with weighted edge connections to neighboring pixels. The authors then formulate panoptic segmentation as a graph cutting problem.
To solve the graph cutting problem, the authors use an algorithm called asymmetric multi-way cut (AMWC). The authors use work from [1] to backpropagate through the AMWC layer, and thus making their architecture end-to-end trainable.
Finally, the authors propose a new differentiable loss function based on the panoptic quality metric.

Overall, the paper is well written and the authors do a good job discussing the limitations and benefits of their work.

Major Cons (please address this in your rebuttal):
The authors did not include any code with their submission, they claim they will release their code upon acceptance.

1) In figure 2 what exactly is the decoder in the affinity branch? This is neither explained in the main body of the paper nor the appendix.

2) What happens when applying work in [1] directly without the averaging trick used in section 3.3.? Also when averaging what values of lambda are considered? I see you mention N=5 on line 293 but it would be nice to have a more in-depth study of this averaging somewhere, since you claim it's one of the major contributions of the paper.

3) The training routine is not well explained. Are you freezing the ResNet params or are they also trained? Does the end-to-end training happen after the baseline pre-training or are those two separate things? I think the answer is they're not separate but it was slightly unclear to me.

-----------------
Minor Cons (do not address in rebuttal):
1) The function total_edge_similarity(ij) on line 25 of algo 2 in the appendix is a little hard to find. Maybe move it to a separate "algo" or move it up to line 3 of algo 2 (before it is first used).
2) Fix sentence on line 222: "also allows one to also "

---------------

[1] Marin Vlastelica Poganciˇ c, Anselm Paulus, Vit Musil, Georg Martius, and Michal Rolinek. Differentiation of blackbox combinatorial solvers. In International Conference on Learning Representations, 2019.


**Time Spent Reviewing:**

4

---

> ### Author Response · Authors · 2021-08-08
> **Rebuttal**
>
> Thank you for your review, positive appraisal of our paper and pointing out minor inconsistencies. We will improve our paper based on raised concerns.
>
> * **Code Availability:** Available via [github](https://github.com/anobutterfly/COPS).
>
> * **Affinity branch:** Thanks for pointing this out. Each semantic and affinity branch uses a DeepLabv3+ decoder [11] similar to Panoptic DeepLab [13]. We will make this clear in the final manuscript.
>
> * **What happens when applying the Vlastelica et al 2019 approach without the averaging trick:** Convergence becomes slower (as per number of iterations and wall clock time) and the reached accuracy becomes lower. We have an exemplary convergence plot with and without averaging in Figure 4 for Cityscapes ($N=1$ corresponds to Vlastelica et al 2019). We observed the same for COCO. Such improvement can be explained by the following:
> If we directly use the approach of Vlastelica et al 2019 the gradient can only take values in $\\{-1 / \lambda, 0, 1/\lambda \\}$. Thus the gradient is giving a direction of loss decrease/increase but without any differences in the absolute value. By our robust backpropagation approach, the gradients get averaged and thus they become more informative, including different absolute values.
>
>     We have created a gif [animation](https://imgur.com/a/zKzFLAr) showing the original image and gradients w.r.t. the semantic branch with different numbers $N$ of $\lambda$-values. What we see is that for single pertubations ($N=1$) the magnitude of the gradient is either zero or $|1/\lambda|$, while our robust backpropagation with multiple $\lambda$ results in a gradient with multiple magnitude levels (shown by varyings shades of grey). This, we argue, helps in driving the neural network weights by larger updates whenever the loss indicates that severe errors are made and by smaller updates when loss is less sensitive w.r.t. the errors made with the current weights.
>
>     We will add more information about this comparison in the final manuscript.
>
> * **Choice of values $\lambda$:** Instead of finding an appropiate value we sample uniformly in a range $[1, 5000]$ for each sample in the batch as shown in Table 2 of the appendix. We will clarify in the main part of the paper.
>
> * **Explanation of training routine, freezing of ResNet parameters:** The end-to-end training happens after the pre-training phase using the panoptic quality surrogate loss. During this stage only the last layer of the semantic branch and the last two layers of the affinity branch are trained. All other parameters are frozen.
> We currently avoid training the whole model with end-to-end training for the following reason: IoU based metrics including the PQ quality (and our surrogate) are not separable under expectations  w.r.t. different images (see eq. 15 in [Ma18]). To get good estimates of the loss we therefore require many images, i.e. larger batch sizes for end-to-end training than for pre-end training. This restriction makes it infeasible to train all layers due to GPU memory limitations. We will make this clear in the final manuscript.
>
> [Ma18] - Berman et. al. The lovász-softmax loss: A tractable surrogate for the optimization of the intersection-over-union measure in neural networks. CVPR 2018.

---

> > ### Comment · Reviewer_UruC · 2021-08-20
> > **Response**
> >
> > Thank you for your comment. After reading your rebuttal and other reviews, I have decided to keep my score at a 6. Other reviewers point out that it is unclear how practical the proposed approach is. I find that at least parts of the proposed methodology is of practical interest to the community.

---

### Official Review · Reviewer_aCGb · 2021-07-22

**Rating:** 3
**Confidence:** 5

**Summary:**

This paper proposed a new architecture for panoptic segmentation consisting of a convolutional neural network and an asymmetric multiway cut problem solver.
Their formulation allows to directly maximize a smooth surrogate of the panoptic quality metric by backpropagating the gradient through the optimization problem.

**Limitations And Societal Impact:**

-- It is unclear what is the advantages of the proposed method.
-- There is not ablation study of the new modules.

**Main Review:**

-- Overclaim. The paper claims the proposed method is 'An End-to-End Trainable Approach', however, it requires a pre-training for semantic and affinity branches, which is followed by fine-tuning the overall model.

-- Performance. The proposed approach is complex and does not achieve the state-of-the-art results.


**Time Spent Reviewing:**

4

---

> ### Author Response · Authors · 2021-08-08
> **Rebuttal**
>
> Thank you for your review and the raised concerns. We will better address raised criticism in a final version of our paper.
>
> * **Complexity:** We argue that our pipeline is in fact on the simpler side of existing panoptic segmentation approaches. As is common, we have a backbone and a semantic and affinity branch in our neural network followed by a single optimization problem producing the final segmentation. Other works have different postprocessing schemes that are at least as complex as our AMWC step:
>   1. For example Panoptic Deeplab [13] uses a handcrafted voting scheme based on center probabilities and offsets. This voting scheme needs non-maximum suppression and probability thresholding, introducing more components in their approach and making it non-differentiable.
>   2. For top-down approaches like Max-Deeplab [60] the architectures are more complex than ours. [60] additionally includes transformers that need probability thresholds and a preset maximum number of instances the architecture can detect.
>   3. Other top-down approaches like UPSNet [68] additionally require a Mask-RCNN and use a total of 8 loss terms for training which introduces more hyperparameters.
>   4. The work [23] uses multiple calls to a multicut solvers in a coarse-to-fine manner and fuses the produced results in comparison to a single call to an AMWC solver for our work.
>
>   Please see Table 3 in the supplementary material for detailed comparison in terms of hyperparameters.
>
> * **End-to-End training overclaim vs pre-training & fine-tuning:** See our answer in the overall comment section.
>
> * **Performance:** We outperform comparable bottom-up baselines by significant margins w.r.t. Panoptic Quality. Only significantly more complex models using top-down information, several times larger backbones and/or more data augmentation achieve superior performance to our approach. The goal of our paper was to improve upon existing lightweight architectures.
>
> * **Ablation study:** We provide several ablations. We compare against a baseline with pre-training but without end-to-end training (see Table 1) and end-to-end training without the PQ surrogate loss (Table 4 in the appendix). We also show that non-robust backpropagation(i.e. single $\lambda$) gives worse convergence (Figure 4). In summary, every contribution (end-to-end training, PQ surrogate, robust backpropagation) is necessary for improving upon the baseline.

---

### Author Response · Authors · 2021-08-08
**General Rebuttal**

We thank the reviewers for their thoughtful and informative reviews. Some general points raised by multiple reviewers are addressed below:

* **Inference time bottleneck (upv8, G2zw):** We agree that the inference speed is not yet satisfactory as we also pointed out in our conclusion. However, our approach will benefit from future advances in two regimes:
    1. _Asymmmetric multiway cut methods:_ We formulate panoptic segmentation as an optimization problem in contrast to ad-hoc methods that use hand-crafted heuristics. We hope that we will make the AMWC problem more prominent and this in turn will stimulate investigations into efficient algorithms. However, efficient algorithm design for AMWC is beyond the scope of this paper.
    2. _Learning to adapt graph structure:_ Our AMWC problem instances are created on a fixed graph. Investigation into methods of adapting the graph structure w.r.t scene content can possibly lead to decreased run-time. In our work we focused on conceptual improvements and simplicity of the pipeline. We think that learning to adapt to the graph structure is an orthogonal research direction to ours and therefore we did not include such potential improvements to our pipeline.

* **Two-stage training (aCGB, 8yas, upv8, G2zw):** Pre-training and subsequent end-to-end fine-tuning is common and exactly this setting is called end-to-end training, see e.g. [He21], [55] and [56]. We follow this two-step approach due to computational efficiency, since the combinatorial part takes a significant amount of time. For panoptic segmentation approaches all other methods, to our knowledge, pretrain on Imagenet and then finetune _without_ end-to-end training on the respective dataset. We agree that currently two-stage training is necessary due to computational constraints induced by the AMWC solver. We hope that with better and faster AMWC solvers training can be converted to a single stage in the future. However, our comparison in Table 1 shows the benefit of our end-to-end training phase in comparison to the baseline.


[He21] He, Jiawei, et al. Learnable Graph Matching: "Incorporating Graph Partitioning with Deep Feature Learning for Multiple Object Tracking" in CVPR 2021

---

### Decision · Program_Chairs · 2021-09-28

**Decision:**

Accept (Poster)

**Comment:**

This work introduces a fully differentiable method for training a model for panoptic image segmentation. This method consists of two parts: (1) constructing and training an approximate proxy metric for PQ (panoptic quality), (2) posing the problem in terms of affinity prediction between adjacent pixels and subsequently using an asymmetric multiway cut solver (AMWC) to arrive at differentiable prediction of panoptic segmentation. The model is trained in 2 stages: first the model is pre-trained using the proxy metric, second the model is fine-tuned using the AMWC solver. The quality of the model is subsequently evaluated on COCO and Cityscapes datasets and compares favorably to several baselines.

Although there was some notable spread across the reviews, in general the reviewers commented positively on the novelty of the proposed combinatorial optimization and panoptic proxy loss. Although the inference speed of the optimization method is quite slow, the method does replace lots of hand-crafted heuristics. Otherwise, the reviewers did raise significant concerns about the presentation quality of the paper, the need for providing additional ablations, and questions about the comparison to baselines. In my read of the paper I was surprised how unfavorably the proposed method performs with respect to some of the reported baselines (e.g. Table 1, Axial-DL [60] 42.2 vs. 36.6); likewise, I saw instances of heuristics that made me question whether hand-crafted heuristics were removed (e.g. line 256). Finally, I was additionally concerned whether the paper was fully self-contained as important ablations are contained within the Appendix. All of these issues appear substantial and raise the general concern that the amount of updates necessary for acceptance would be quite significant and require further review.

One item of note that the reviewers highlighted was the claim that the model was "end-to-end trained" and whether this claim matches the fact that the model requires a pre-training step followed by fine tuning. The necessity of the two-stage training procedure is due to the slow inference time of the proposed AMWC solver, and thus fundamental to the proposed method. Although there is some slight confusion in the literature about the strictness of using the term "end-to-end training", I do concur with the reviewers that the term "end-to-end" is intended to imply that a single optimization/training stage is sufficient for training. Furthermore, after having consulted with multiple colleagues, I believe that this is the general sentiment of the larger deep learning community. I would suggest that the authors employ their suggestion of "fully differentiable" in subsequent revisions as many reviewers would find the current title inappropriate for describing the methodology.

In summary, the work offers a promising idea that presents a novel method for training a model to perform panoptic segmentation. That said, the reviewers still offer several strong concerns about the presentation and the baselines that require substantial revisions by the authors. In sum, this work is borderline, and given that there is no strong champion for this work to be accepted, this paper will not be accepted at NeurIPS. I strongly suggest that the authors heavily revise their manuscript and resubmit to the next available conference, potentially with a computer vision focus.


**Consistency Experiment:**

NeurIPS has a long history of experimentation. In 2014, NeurIPS ran an experiment in which 10% of submissions were reviewed by two independent committees to quantify the randomness in the review process. This year, we repeated a variant of this experiment to see how the quality of the review process has changed over time.  This paper was part of the experiment and was therefore assigned to two committees (consisting of reviewers, an Area Chair, and a Senior Area Chair) that reached independent decisions.  If both committees made the same recommendation, this recommendation was followed. If a single committee recommended acceptance, the paper was accepted (with the exception of a few cases in which the other committee identified what we considered a fatal flaw, e.g., an error in a key result).

This copy’s committee reached the following decision: **Reject**

The other committee assigned to the paper recommended **Accept (Poster)**.  You can find the other set of reviews, along with any follow up discussion with the authors here:
https://openreview.net/forum?id=70eD741FHyI